evolution/psychology

sexual unfaithfulness, accuracy, first impressions

**Author for correspondence:**
Yong Zhi Foo
e-mail: yong.foo@uwa.edu.au

# Sexual unfaithfulness can be judged with some accuracy from men's but not women's faces

Yong Zhi Foo[1,2], Antonina Loncarevic[1], Leigh W. Simmons[1,2], Clare A. M. Sutherland[1] and Gillian Rhodes[1]

[1]ARC Centre of Excellence in Cognition and its Disorders, School of Psychological Science, and [2]Centre for Evolutionary Biology, School of Biological Sciences, University of Western Australia, 35 Stirling Hwy, Crawley, 6009 Western Australia, Australia

(iD) YZF, 0000-0001-7627-2991; LWS, 0000-0003-0562-1474

We routinely make judgements of trustworthiness from the faces of others. However, the accuracy of such judgements remains contentious. An important context for trustworthiness judgements is sexual unfaithfulness. Accuracy in sexual unfaithfulness judgements may be adaptive for avoiding reproductive costs associated with having an unfaithful partner. Indeed, emerging studies suggest that women, and to a lesser degree, men, show above-chance accuracy in judging sexual unfaithfulness from opposite-sex faces. In the context of mate guarding, it is important not only to assess the likelihood of a partner defecting, but also to detect same-sex poachers. Therefore, here, we examine whether individuals can also judge sexual unfaithfulness (self-reported cheating and poaching behaviour) from same-sex faces. We found above-chance accuracy in judgements of unfaithfulness from same-sex faces in men but not women. Conversely, we found above-chance accuracy for opposite-sex faces in women but not men. Therefore, both men and women showed above-chance accuracy, but only for men's, and not women's, faces. Raters were making accurate (above-chance) judgements of unfaithfulness from men's faces using facial masculinity, a well-established signal of propensity to adopt short-term mating strategies. In summary, we found above-chance accuracy in impressions of unfaithfulness from men's faces. Although very modest, the level of accuracy could nevertheless have biological significance as an evolved adaptation for identifying potential cheaters/poachers.

# 1. Unfaithfulness can be judged with some accuracy from men's but not women's faces

Judging trustworthiness from faces is a fundamental social phenomenon. Trustworthiness represents one of the core dimensions underlying trait judgements that are commonly made from faces [1–3]. People show substantial reliability and consensus with others in their judgements (e.g. [4,5]). They also make such judgements rapidly [6–8] and spontaneously [9,10]. The importance of trustworthiness judgements is further demonstrated by their potential impact on critical social outcomes. Untrustworthy-looking individuals are less likely to be trusted in economic trust game settings (e.g. [11–13]). They are also more likely to be judged as guilty in a simulated court setting despite evidence to the contrary [14]. Yet despite the level of consensus and automaticity at which we form such a core trait judgement and its potential social consequences, the accuracy of these judgements remains contentious [5,15]. Indeed, studies on the accuracy of trustworthiness judgements using both dispositional (e.g. general honesty) and domain-specific (e.g. being convicted of a crime) behavioural measures of actual trustworthiness have found mixed results [4,5,16].

One domain of trustworthiness that is of particular interest is sexual unfaithfulness. Humans are characterized by long-term pair-bonds, in which both sexes invest substantially in their partners and offspring for extended periods of time [17,18]. From an evolutionary perspective, there are significant reproductive costs associated with having an unfaithful partner. Both sexes risk losing valuable resources or even one's mate to a competitor [17–19]. In addition, men also risk being cuckolded and investing their resources in a genetically unrelated child [17–19]. It is unsurprising, therefore, that sexual unfaithfulness is one of the strongest factors in the maintenance of pair-bonds. In a study across 160 cultures, including both industrialized and non-industrialized societies, infidelity was the most commonly cited cause of divorce [20]. Given the reproductive costs of being cheated on, evolutionary theories predict that it would be adaptive for individuals to evolve strategies to prevent sexual infidelity [19,21]. Accuracy in judging sexual unfaithfulness of others might, therefore, represent one such strategy. In this context, judgements of the propensity for sexual unfaithfulness made from the faces of strangers could play an important role in reducing the risk of developing relationships with partners who may prove unfaithful.

The face plays an important role in human mate choice as a signal of various aspects of quality, including genetic quality [22,23], diet [24], fertility [25–27], aggressiveness [28] and parental care [29,30]. Recent studies suggest that our faces might also provide signals to unfaithfulness and that we possess some level of accuracy in judging unfaithfulness from opposite-sex faces [31–33]. Women's ratings of sexual unfaithfulness from men's faces correlated positively and significantly, albeit modestly, with those men's self-reported cheating (number of extra-pair copulation partners) and poaching (number of sexual partners already in a relationship) behaviour [31,33]. No significant accuracy was found for men's ratings from women's faces [31], although men can perform above chance when asked to pick the more unfaithful face from pairs consisting of a self-reported cheater and a self-reported non-cheater [32]. In summary, women and to a lesser degree men, appear to have some modest level of accuracy in judging sexual unfaithfulness from opposite-sex faces.

The studies have also examined the facial cues driving accuracy in unfaithfulness judgements, focusing on two potentially valid cues, namely attractiveness and sexual dimorphism. There are substantive reasons for linking these two cues to actual unfaithfulness. Attractive individuals are preferred as sexual partners and are subject to more attempts by the opposite sex to lure them into extra-pair relationships [34]. Sexual dimorphism, particularly male masculinity, is positively related to preference for uncommitted sex and multiple matings [35]. For men's faces, masculinity mediated the relationship between perceived unfaithfulness rated by women and actual unfaithfulness, indicating that women used the valid cue of masculinity to assess men's sexual unfaithfulness at above-chance levels [31,33]. For women's faces, even though attractiveness and femininity were related to perceived infidelity, neither cue was related to actual infidelity [32]. Therefore, it remains unclear what cues might be driving accuracy in men's judgements of women's unfaithfulness.

So far, studies on the potential accuracy of unfaithfulness judgements from faces have considered only opposite-sex judgements. Here, we ask whether there is any accuracy in judgements of unfaithfulness from same-sex faces. There are good theoretical reasons to think that there would be. Poaching is a common mating strategy across cultures [31,34,36,37]. In a cross-cultural ethnographic survey, extra-marital sex was found to be 'not uncommon' in 33 out of 56 cultures (58.9%) for women and 38 out of 55 cultures (69.1%) for men [36]. Up to 70% of individuals across more than 50 cultures

report having attempted to poach someone else's partner before [34,37]. Within these cultures, up to 60% of individuals report succeeding at poaching someone's partner at least once [31,34,37]. Given the prevalence of mate poaching, being able to identify and deter same-sex rivals is likely to be another important factor in determining the success of one's mate-guarding efforts. Indeed, both men and women report engaging in various behavioural strategies that serve to deter same-sex poachers. Strategies include signalling to potential rivals that the partner is already taken (e.g. holding the partner's hand when others are around or requesting the partner to wear ornaments that signify possession), threatening potential rivals, or using physical violence to drive off potential rivals [21,38].

In the context of deterring same-sex poachers, therefore, there might be selection pressure for some level of accuracy in judging unfaithfulness of same-sex strangers because it would help us identify potential rivals. Hence, we examine whether people show above-chance accuracy in judging sexual unfaithfulness from same-sex faces. We also examine people's accuracy in judging opposite-sex faces as a replication of previous findings using the same face database but with a new set of participants. Previous studies on accuracy of unfaithfulness judgements have focused primarily on group-level accuracy (i.e. ratings of unfaithfulness averaged across raters; see [33] for a recent exception). Here, we take the same approach as [33] by testing both group-level and individual-rater-level accuracy. We examine two potential cues, attractiveness and sexual dimorphism, to these impressions and any accuracy observed. We also ask whether any accuracy observed is specific to unfaithfulness impressions and not just general impressions of trustworthiness.

## 2. Method

### 2.1. Participants

We recruited 1516 self-reported heterosexual adult Caucasians (592 men, mean age = 37.4, s.d. age = 12.8, range = 18–75 years; 924 women, mean age = 38.1, s.d. age = 12.8, range = 18–98 years) online from Amazon Mechanical Turk for a sexual unfaithfulness rating study. We aimed for a sample size that was more than 2.5 times (as recommended by Simonsohn [39]) the number of raters in the original study on accuracy in unfaithfulness judgements [31].

### 2.2. Material

Front-view, colour photographs of faces with neutral expressions of 189 Caucasian adults (101 men and 88 women) were taken from [40]. These are the same faces used in previous unfaithfulness judgement accuracy studies [31–33]. A black oval mask covered most of the hair, neck and ears of the faces. Four additional faces (two men and two women) were used in the practice trials. Self-reported cheating and poaching data for these individuals, and rated attractiveness, sexual dimorphism and untrustworthiness of their faces were also taken from [40]. Attractiveness and sexual dimorphism were originally rated on a seven-point scale (1 = Not attractive/masculine or feminine, 7 = Very attractive/masculine or feminine). Untrustworthiness was originally rated on a 10-point scale (1 = Not very likely to be untrustworthy, 10 = Very likely to be untrustworthy).

### 2.3. Procedure

Participants were randomly assigned to rate the sexual unfaithfulness (How likely is this person to be unfaithful?) of either the men's or women's faces on a 10-point Likert scale (1 = Not at all likely, 10 = Extremely likely). Participants completed two practice trials prior to the ratings task. Faces were presented in a random order and each face remained on screen until the participant responded. A total of 293 men and 472 women rated the women's faces and 299 men and 452 women rated the men's faces.

## 3. Results

### 3.1. Accuracy of unfaithfulness judgements

We initially tested whether ratings of unfaithfulness predicted the self-reported cheating and poaching of each face identity using generalized linear mixed models with both face ID and rater ID as random

factors. Because cheating and poaching scores were count data with a Poisson-like distribution, we ran our analyses with a negative binomial distribution. We used R [41] package glmmTMB [42]. In accordance with [43], we first specified the maximal random effects structure and successively reduced its complexity until we achieved convergence. The model that achieved convergence included only the random intercept of rater ID. According to the simulations of [43], random-intercept-only mixed-effect models have a higher Type-1 error rate compared to the traditional approach of analysing the data at the group level (i.e. averaging ratings across raters for each face) and then examining whether the results are also significant at the individual-rater level [44]. Therefore, we discarded the GLMM approach and adopted the traditional approach of analysing data at both the group and rater levels.

### 3.1.1. Group-level accuracy

There was high reliability in the sexual unfaithfulness ratings at the group level for both sexes of face and both sexes of raters as shown by the high Cronbach's alpha levels (men's faces: $alpha_{female\ raters} = 0.98$, $alpha_{male\ raters} = 0.96$; women's faces: $alpha_{female\ raters} = 0.98$, $alpha_{male\ raters} = 0.97$). The high reliability allowed us to analyse the data using average unfaithfulness rating for each face. Average ratings were calculated for both men's and women's faces, separately for each sex of rater. The descriptive statistics of the group-level rated unfaithfulness, other facial impression ratings, age and self-reported cheating and poaching of the faces of both sexes are presented in table 1.

We tested whether average rated unfaithfulness predicted actual cheating and poaching scores using generalized linear models with a negative binomial distribution. Both men's and women's average rated unfaithfulness ratings positively predicted the cheating and poaching scores of men's faces (table 2). However, neither predicted these scores for women's faces (table 2). Therefore, both men and women were accurate in assessing men's, but not women's, likelihood to cheat and poach.

For completeness, we also presented our group-level results in the same form as previous studies [31,32]. Zero-order correlations between age, cheating and poaching scores, and the various facial impression ratings are reported in electronic supplementary material, tables S1 and S2.

### 3.1.2. Rater-level accuracy

Individuals can vary in their unfaithfulness ratings even when there is high consensus at the group level. To assess whether the unfaithfulness ratings were accurate at the rater level, we used each individual's ratings to predict cheating and poaching using negative binomial regression models (figures 1 and 2). The regression slopes provide a measure of accuracy. The individual rater-level negative binomial regression model failed to converge and provide an estimate of the regression slope in 11 instances (two men and three women predicting cheating men's faces, one man and one woman predicting cheating in women's faces, one man and three women predicting women's poaching). These instances were treated as missing data (see table 4 for final d.f.). The percentage of raters who showed above-chance accuracy ranged from 14.1 to 18.0% for judgements of men's faces and from 0.9 to 4.0% for women's faces (table 3).

One-sample *t*-tests (comparing accuracy to zero) indicated that for men's faces, both men's and women's unfaithfulness ratings showed significant accuracy at the individual-rater level for both cheating and poaching (table 4). For women's faces, there was no evidence of accuracy at the individual-rater level except for men's ratings predicting women's poaching (table 4).

## 3.2. Cues to perceived unfaithfulness

Given that the average unfaithfulness ratings were very highly correlated across rater sex (0.96 for men's faces and 0.94 for women's faces; electronic supplementary material, tables S1 and S2), we ran all subsequent analyses using a single rated unfaithfulness score for each face averaged across the male and female ratings. To examine the cues used to judge unfaithfulness, for men's faces, we entered age of the model, sexual dimorphism, attractiveness and untrustworthiness into a general linear regression model to examine which of them predicted rated unfaithfulness. For women's faces, the same analyses were conducted with the exception that sexual dimorphism was excluded from the analyses due to its high correlation, and therefore potential multicollinearity, with attractiveness (electronic supplementary material, table S2).

**Table 1.** Descriptive statistics for age, self-reported cheating and poaching, and facial impression ratings for both sexes of faces. Average untrustworthiness, attractiveness and sexual dimorphism ratings were taken from [40] (attractiveness and sexual dimorphism ratings missing for three men's faces).

| | men's faces | | | | | women's faces | | | | |
|---|---|---|---|---|---|---|---|---|---|---|
| | n | mean | s.d. | skew | kurtosis | n | mean | s.d. | skew | kurtosis |
| age | 101 | 24.6 | 6.9 | 1.55 | 1.65 | 88 | 24.4 | 5.8 | 1.55 | 2.38 |
| cheating | 101 | 1.4 | 3.9 | 5.03 | 29.88 | 88 | 0.7 | 1.9 | 5.56 | 38.09 |
| poaching | 101 | 0.6 | 1.3 | 3.41 | 13.16 | 88 | 0.4 | 0.7 | 1.48 | 1.48 |
| unfaithfulness rated by men | 101 | 5.4 | 0.5 | −0.14 | 0.21 | 88 | 4.5 | 0.5 | 0.32 | −0.13 |
| unfaithfulness rated by women | 101 | 5.3 | 0.6 | −0.06 | −0.01 | 88 | 4.5 | 0.6 | 0.24 | −0.43 |
| untrustworthiness | 101 | 5.8 | 0.8 | 0.07 | −0.21 | 88 | 5.6 | 0.7 | −0.42 | −0.38 |
| attractiveness | 98 | 2.9 | 0.9 | 0.26 | −0.33 | 88 | 2.9 | 0.9 | 1.12 | 2.03 |
| sexual dimorphism | 98 | 4.5 | 0.9 | −0.15 | −0.75 | 88 | 3.9 | 1.0 | 0.45 | −0.24 |

**Table 2.** Results of negative binomial generalized linear models of ratings of unfaithfulness predicting cheating and poaching scores for men's faces.

| | cheat | | | | | poach | | | | |
|---|---|---|---|---|---|---|---|---|---|---|
| | estimate | s.e. | z | p | $R^2$ | estimate | s.e. | z | P | $R^2$ |
| *men's faces* | | | | | | | | | | |
| Model 1 | | | | | | | | | | |
| unfaithfulness rated by men | 1.01 | 0.46 | 2.17 | 0.03 | 0.04 | 0.75 | 0.37 | 2.04 | 0.04 | 0.04 |
| Model 2 | | | | | | | | | | |
| unfaithfulness rated by women | 1.01 | 0.38 | 2.65 | 0.01 | 0.06 | 0.91 | 0.31 | 3.00 | 0.00 | 0.08 |
| *women's faces* | | | | | | | | | | |
| Model 1 | | | | | | | | | | |
| unfaithfulness rated by men | 0.15 | 0.46 | 0.33 | 0.74 | 0.00 | 0.14 | 0.32 | 0.44 | 0.66 | 0.00 |
| Model 2 | | | | | | | | | | |
| unfaithfulness rated by women | 0.08 | 0.38 | 0.21 | 0.83 | 0.00 | 0.12 | 0.27 | 0.46 | 0.65 | 0.00 |

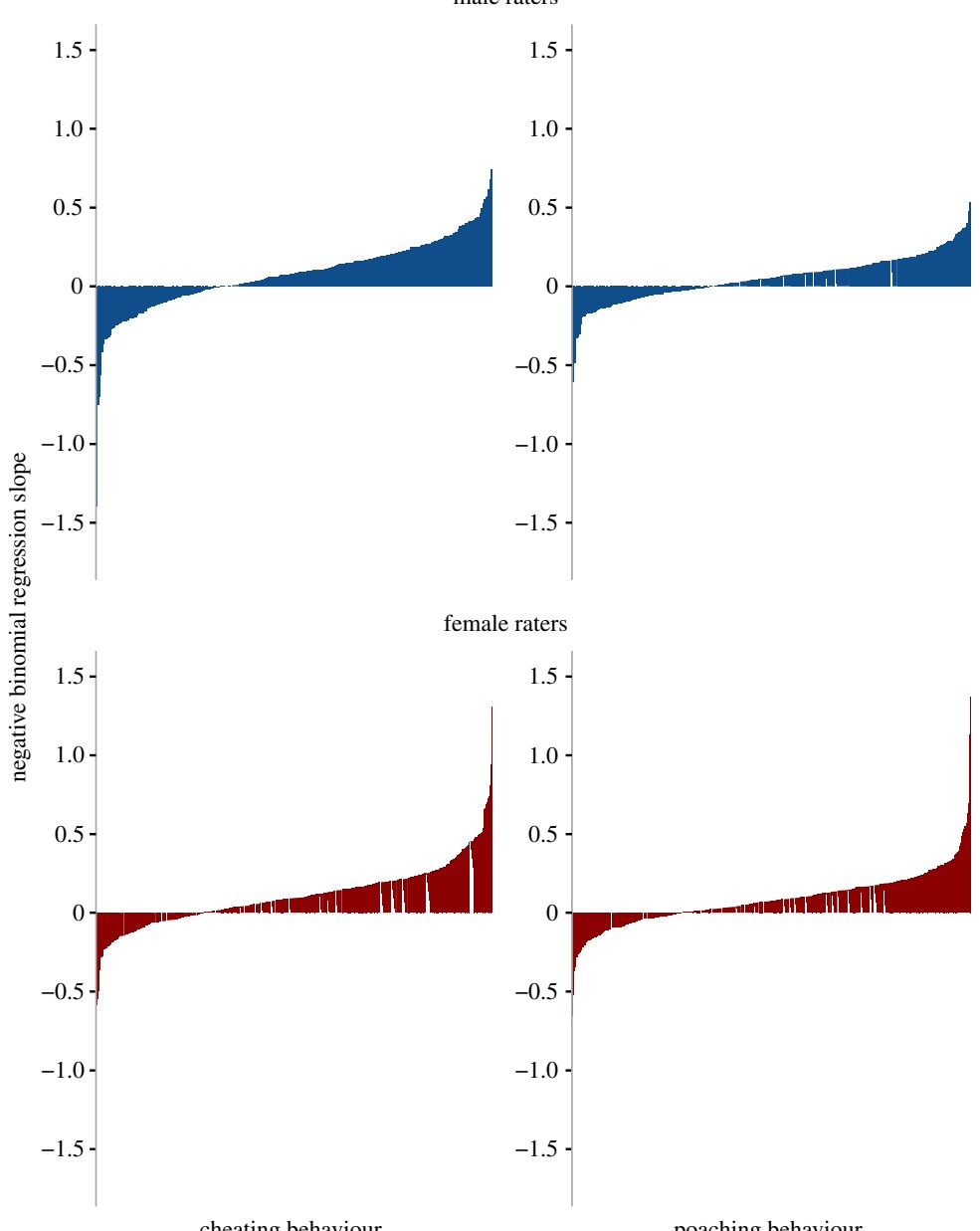

**Figure 1.** Individual-rater accuracy in predicting cheating and poaching behaviour from men's faces. Each rater's accuracy is represented by one vertical line.

For men's faces, rated unfaithfulness was positively predicted by sexual dimorphism, attractiveness and untrustworthiness (table 5). For women's faces, rated unfaithfulness was positively predicted by attractiveness and untrustworthiness (table 5).

## 3.3. Valid cues to unfaithfulness in men's faces

Our next step was to identify which of the cues that were used to judge unfaithfulness from men's faces, namely sexual dimorphism, attractiveness and untrustworthiness, were valid cues. Note that, to be valid, the cue must predict cheating/poaching behaviour in the same direction as it predicts rated unfaithfulness. We entered sexual dimorphism, attractiveness and untrustworthiness simultaneously into two negative binomial generalized linear regression models to predict men's cheating and poaching scores, respectively. Both cheating and poaching scores were positively predicted by sexual dimorphism (table 6). Surprisingly, even though more attractive men were rated as more unfaithful (table 5), they were less likely to engage in actual mate poaching (table 6).

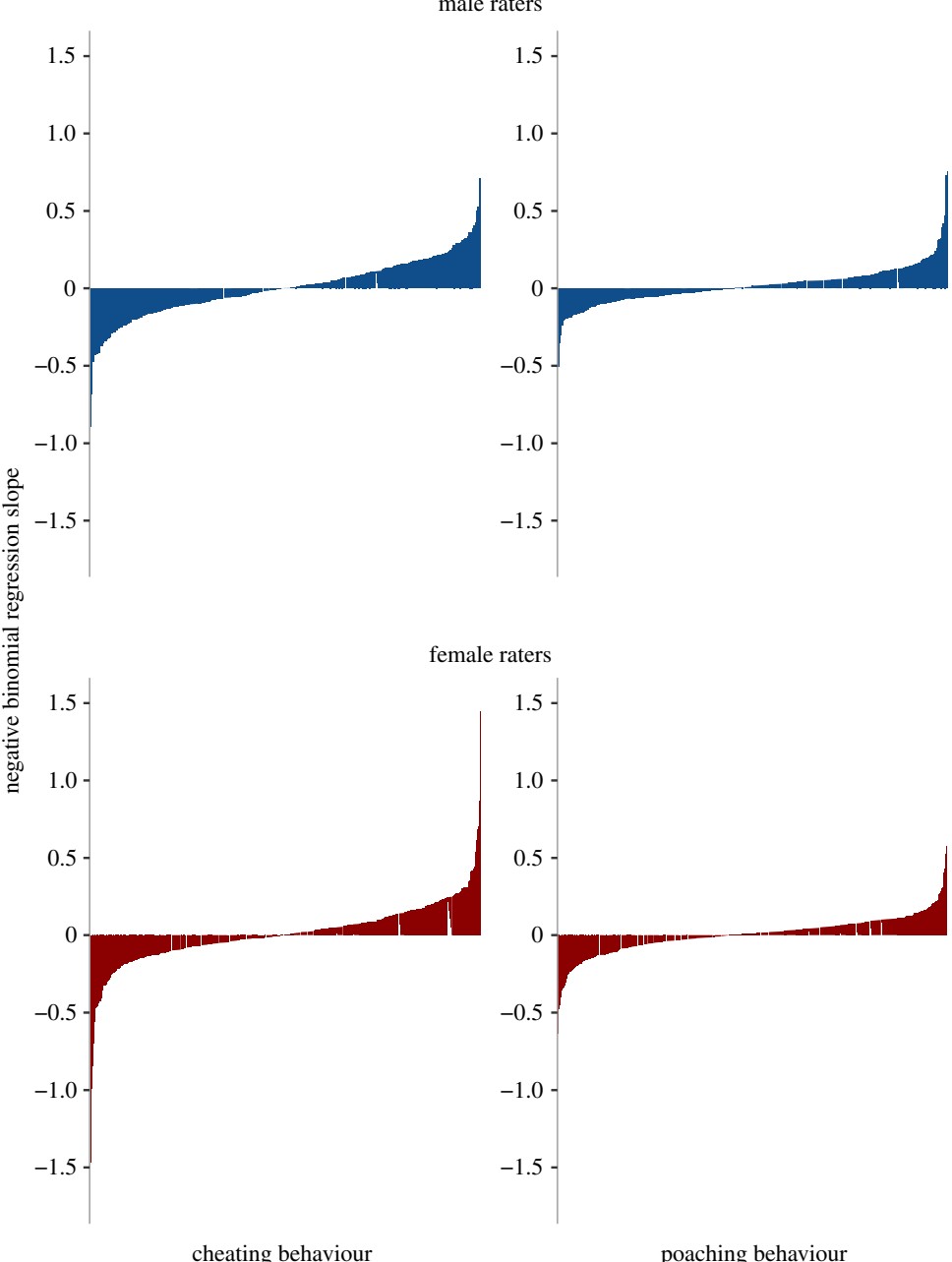

**Figure 2.** Individual-rater accuracy in predicting cheating and poaching behaviour from women's faces. Each rater's accuracy is represented by one vertical line.

**Table 3.** Percentage of raters who showed above-chance individual accuracy and their mean and s.d. accuracy by sex of face and sex of rater.

| | cheating | | poaching | |
|---|---|---|---|---|
| | % accurate raters | mean (s.d.) accuracy | % accurate raters | mean (s.d.) accuracy |
| men's faces | | | | |
| men | 14.1 | 0.37 (0.13) | 12.0 | 0.27 (0.09) |
| women | 16.6 | 0.34 (0.16) | 18.0 | 0.26 (0.11) |
| women's faces | | | | |
| men | 4.0 | 0.33 (0.14) | 3.7 | 0.31 (0.23) |
| women | 3.3 | 0.42 (0.35) | 0.9 | 0.37 (0.16) |

**Table 4.** Results of one-sample *t*-tests for above-zero individual accuracy (individual regression slopes).

| | cheating | | | | | poaching | | | | |
|---|---|---|---|---|---|---|---|---|---|---|
| | accuracy | s.d. | *t* | d.f. | *p* | accuracy | s.d. | *t* | d.f. | *p* |
| men's faces | | | | | | | | | | |
| men | 0.08 | 0.23 | 5.71 | 290 | 0.00 | 0.05 | 0.15 | 6.00 | 292 | 0.00 |
| women | 0.11 | 0.20 | 11.62 | 468 | 0.00 | 0.08 | 0.18 | 10.41 | 471 | 0.00 |
| women's faces | | | | | | | | | | |
| men | 0.00 | 0.19 | 0.04 | 297 | 0.97 | 0.02 | 0.13 | 2.43 | 297 | 0.02 |
| women | 0.01 | 0.22 | 0.74 | 450 | 0.46 | 0.00 | 0.12 | 0.75 | 448 | 0.45 |

**Table 5.** Results of general linear regression models testing the cues that were used to judge unfaithfulness from men's and women's faces.

| | men's faces | | | | women's faces | | | |
|---|---|---|---|---|---|---|---|---|
| | *B* | s.e. | *t* | *p* | *B* | s.e. | *t* | *p* |
| age | 0.01 | 0.01 | 1.28 | 0.20 | −0.01 | 0.01 | −1.71 | 0.09 |
| sexual dimorphism | 0.25 | 0.06 | 4.17 | 0.00 | — | — | — | — |
| attractiveness | 0.33 | 0.06 | 5.20 | 0.00 | 0.54 | 0.04 | 13.45 | 0.00 |
| untrustworthiness | 0.29 | 0.06 | 4.74 | 0.00 | 0.37 | 0.05 | 7.66 | 0.00 |

**Table 6.** Results of negative binomial generalized linear models test the cues that provide valid signals to men's cheating and poaching.

| | estimate | s.e. | *z* | *p* |
|---|---|---|---|---|
| Model 1: predicting men's cheating scores | | | | |
| sexual dimorphism | 0.77 | 0.29 | 2.67 | 0.01 |
| attractiveness | −0.49 | 0.31 | −1.56 | 0.12 |
| trustworthiness | −0.49 | 0.34 | −1.44 | 0.15 |
| Model 2: predicting men's poaching scores | | | | |
| sexual dimorphism | 0.49 | 0.22 | 2.20 | 0.03 |
| attractiveness | −0.64 | 0.26 | −2.48 | 0.01 |
| trustworthiness | −0.27 | 0.27 | −1.01 | 0.31 |

## 3.4. Driver of accuracy in men's faces

Given that sexual dimorphism was positively related to both rated unfaithfulness and actual cheating and poaching scores, we entered sexual dimorphism together with unfaithfulness ratings into the negative binomial generalized linear regression models predicting men's cheating and poaching scores. The aim was to test whether sexual dimorphism accounted for the relationships between rated unfaithfulness and cheating and poaching scores reported in table 2. If sexual dimorphism was used by raters to make valid judgements of sexual unfaithfulness, then entering sexual dimorphism together with rated unfaithfulness would reduce the relationship between rated unfaithfulness and actual infidelity. Both rated unfaithfulness and sexual dimorphism became non-significant in all regression models (table 7). Therefore, sexual dimorphism was used by raters as a valid cue to assess cheating and poaching behaviour in men.

**Table 7.** Results of negative binomial generalized linear models showing that sexual dimorphism accounted for the relationship between ratings of unfaithfulness and actual infidelity, indicating that sexual dimorphism is a driver to accuracy in men's faces.

|  | estimate | s.e. | z | p |
|---|---|---|---|---|
| **Model 1: predicting men's cheating scores** | | | | |
| rated unfaithfulness | 0.69 | 0.54 | 1.27 | 0.20 |
| sexual dimorphism | 0.46 | 0.35 | 1.33 | 0.18 |
| **Model 2: predicting men's poaching scores** | | | | |
| rated unfaithfulness | 0.70 | 0.44 | 1.56 | 0.12 |
| sexual dimorphism | 0.22 | 0.29 | 0.78 | 0.43 |

## 4. Discussion

We found above-chance accuracy in unfaithfulness judgements of same-sex faces, but only for men rating men's faces and not women rating women's faces. Our results were not as expected. Given the reproductive costs to having an unfaithful partner and the prevalence of mating poaching attempts [17–19], we expected some level of accuracy in judging unfaithfulness of same-sex strangers for mate-guarding purposes in both sexes of raters. However, we found above-chance accuracy only in men's ratings. Using the same face database but with a new set of participants, we also replicated previous findings of above-chance accuracy for opposite-sex faces, but only for women rating men's faces and not men rating women's faces. Taken together, both men and women showed above-chance accuracy for men's faces but not women's faces. Therefore, perceived unfaithfulness may indeed contain some kernel of truth in male faces [31].

The original work on accuracy of rated unfaithfulness for men's faces has examined only group-level accuracy [31]. Here, similar to the recent findings by Sutherland *et al.* [33], we found accuracy for men's faces not only at the group level, where there were above-chance relationships between average rated unfaithfulness and actual infidelity across face identities, but also at the rater level, where the average rater accuracy score was significantly higher than zero. Therefore, the group-level accuracy is not simply an artefact of removing noise at the individual-rater level [45].

Accuracy in men's faces was driven by sexual dimorphism, as found by Rhodes *et al.* [31] and Sutherland *et al.* [33], confirming that sexual dimorphism was used by raters as a valid cue to unfaithfulness. Male masculinity signals men's tendency to adopt short-term mating strategies [46,47], with more masculine men having more sexual partners [40] and having more positive attitudes towards uncommitted sex and multiple matings [35,48]. Therefore, accuracy in judging men's unfaithfulness based on masculinity may represent an evolved adaptation for identifying potential cheaters on the part of female raters and potential poachers on the part of male raters [19,21,31].

Even though accuracy for men's faces was statistically significant, the level of accuracy was modest at best. The percentage variance shared between rated unfaithfulness and actual male infidelity (i.e. cheating and poaching) at the group level ranged from 4 to 8%, which translates to a very modest effect size $r$ of 0.20–0.28. We found similar effect sizes in the rater-level accuracy (0.17–0.23). However, it is important to note that small effects can still have long-term evolutionary consequences. The effect sizes in this study ($r = 0.20$–$0.28$) are comparable to those typically found in evolutionary studies, which are in the range $r = 0.16$–$0.25$ [49]. Even much smaller effects can still have substantial evolutionary impacts if those effects are consistently selected for across multiple generations [49,50]. Therefore, although our effect sizes may seem small by traditional conventions [51], they can still be evolutionarily important.

Our group-level effect sizes are also typical of those from the field of psychology. Indeed, several recent studies have shown that the average effect size in psychology is around $r = 0.2$ [52,53]. Therefore, although the amount of variance shared between rated and actual unfaithfulness is modest, it may still have psychological relevance. Despite the potential evolutionary and psychological relevance at the group level, our results should not, however, be taken to mean that individuals should rely on facial impressions to judge men's unfaithfulness in everyday situations. Researchers have recently cautioned against interpreting statistically significant accuracy as being meaningful for individual diagnosticity [15,33]. Indeed, despite the statistical significance of our group-level results, the small percentage of variance shared between rated and actual unfaithfulness indicates that a large

proportion of variance in actual unfaithfulness remains unaccounted for. Furthermore, although individual-rater-level accuracy was significant on average, only a small percentage of individual raters (14.1–18.0%; table 6) achieved above-chance accuracy in their ratings. Therefore, if a given person were to rely solely on impressions from men's faces to decide who is a cheater or poacher, they would risk substantial error.

There are several explanations for low accuracy of judgements from men's faces. First, although sexual dimorphism was a valid cue, it did not perfectly predict actual infidelity (table 6). Therefore, judgements of unfaithfulness based on male sexual dimorphism were not highly accurate. Second, our results also showed that raters used invalid impressions, such as attractiveness or general trustworthiness, to judge unfaithfulness. Therefore, accuracy might have been compromised by these invalid impressions. Lastly, it is possible that accuracy may have been limited by our use of faces of relatively young individuals, who might be prone to cheating and poaching, but have had limited time and/or opportunities to express those tendencies. The mean and range of the number of times the male individuals in our database had cheated on their partner and/or poached someone else's was relatively low (table 1). Future studies might benefit from using a database of faces with a greater range of cheating and poaching experiences, which might reveal greater accuracy in people's judgements of unfaithfulness.

We found very little evidence of any accuracy in impressions of faithfulness from women's faces. The only above-chance accuracy was in men's rating for women's poaching in our rater-level analyses. Despite statistical significance, this effect was five times smaller than the rater-level accuracy for men's cheating and poaching. Furthermore, we did not find the same result in our face-level analyses. Therefore, it is unlikely to be robust. Our results contrast with previous findings on accuracy in people's judgements of sociosexual attitude (i.e. attitude towards uncommitted sex and multiple mating) from people's faces, which found stronger evidence of accuracy for women's faces than men's instead [35,48]. One potential reason for the difference in the pattern of findings is that Boothroyd *et al.* [35,48] examined attitudes towards uncommitted sexual behaviours while the present study examined self-reported measures of cheating and poaching behaviours.

The finding that both men and women could judge men's, but not women's, faces with some modest level of accuracy offers an alternate interpretation to previous findings. Using the same database of faces, previous studies have examined accuracy of unfaithfulness judgements only in opposite-sex faces and found that men's ratings of women's faces were not accurate, whereas women's ratings of men's faces were [31,32]. Such findings were interpreted as indicating that men are less accurate at judging unfaithfulness than women [31,32]. However, we found that both men and women showed accuracy in judging men's, but not women's, faces, suggesting that it is sex of the face rather than the rater that matters, at least for this database of faces.

There are several possible explanations for the lack of accuracy for women's faces. First, there might not be any valid cues to unfaithfulness in women's faces. Sexual dimorphism was highly related to attractiveness in women's faces. Therefore, unlike in men's faces, for which sexual dimorphism provided a valid cue to unfaithfulness, independent of attractiveness, the relationship between sexual dimorphism, perceived unfaithfulness, and actual infidelity in women's faces might have been confounded by the strong relationship between sexual dimorphism with attractiveness. It is also likely that femininity is less strongly related to actual propensity to cheat/poach in women than masculinity is in men (see electronic supplementary material, tables S1 and S2). Furthermore, women might be more likely to engage in semi-permanent cosmetic enhancements such as shaping their eyebrows or lengthening their eyelashes that might influence their perceived unfaithfulness.

Second, it is possible that we did not find any statistical support of accuracy in judgements of women's faces because of the limited range of self-reported cheating and poaching in our face set. The standard deviation of the women's self-reported poaching behaviour, in particular, was at least half the standard deviation of the men's self-reported cheating and poaching data (table 1). However, we note that the mean and range of the women's self-reported cheating behaviour was comparable to that of the men's poaching data, for which we did find evidence of modest accuracy. Therefore, it is unlikely that the lack of accuracy for women's faces was due solely to the limited variance in the women's self-reported behaviours. Nevertheless, we reiterate our call for future studies to use faces with a greater range of cheating and poaching experiences (e.g. older individuals who have had more opportunities to cheat and poach), which might reveal some level of accuracy in people's judgements of unfaithfulness from women's faces.

In summary, our results suggest that there might be some kernel of truth in impressions of unfaithfulness from men's faces. This above-chance accuracy for men's faces is consistent with the evolutionary prediction that accuracy in our judgements of unfaithfulness from strangers' faces might

represent an evolved adaptation for identifying potential male cheaters/poachers. Our findings also suggest that, contrary to previous findings, men and women are comparable in the accuracy of their unfaithfulness judgements. The small effects, however, also indicate that we should not rely on our first impressions to make diagnostic judgements of unfaithfulness in everyday situations.

Ethics. This research was approved by the Human Ethics Committee at the University of Western Australia (ref. no. RA/4/1/2323). All participants provided informed consent prior to participation.

Data accessibility. The dataset has been uploaded as electronic supplementary material.

Authors' contributions. All authors participated in the conception of the research question and study design. Y.Z.F. programmed the online experiment, collected the data and carried out the data analysis. C.A.M.S. participated in the planning of the data analysis. Y.Z.F. drafted the initial manuscript. All authors contributed to the interpretation of the results and manuscript revisions. All authors gave final approval for publication.

Competing interests. We declare we have no competing interests.

Funding. The study is supported by the ARC Centre of Excellence in Cognition and its Disorders (CE110001021), ARC Professorial Fellowships to L.W.S. (DP110104594) and G.R. (DP0877379), an ARC Discovery Outstanding Researcher Award to G.R. (DP130102300) and an ARC Discovery Early Career Researcher Award to C.A.M.S. (DE190101043).

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
