## [Reviewer comments · Royal Society Open Science]

Review History

RSOS-181552.R0 (Original submission)

Review form: Reviewer 1

Is the manuscript scientifically sound in its present form?

Yes

Are the interpretations and conclusions justified by the results?

Yes

Is the language acceptable?

Yes

Is it clear how to access all supporting data?

Yes

Do you have any ethical concerns with this paper?

No

Have you any concerns about statistical analyses in this paper?

No

Recommendation?

Reject

Comments to the Author(s)

The authors examine the accuracy in sexual unfaithfulness judgments of female and male face done by same and opposite sex raters. Both female and male raters were accurate at judging the likelihood of men to cheat and poach but not the likelihood to be unfaithful by females. These judgement seem to be driven by sexual dimorphism, which appears to be a valid cue of infidelity at least for men. The manuscript is well written and overall convincing. Yet, I have two major concerns, which makes me doubt the importance of these findings. The first one is the null result for female faces. Although, it seems plausible that it is more difficult to find cues to infidelity in women's faces, a more reasonable explanation, which the authors also acknowledge, is the limited variability of the self-reported cheating & poaching behavior in women. As the effect sizes are really small of these findings it seems impossible to find any cues to infidelity in female faces. The second major concern is the importance of these findings, it seems to add very little to the already existing literature – males can judge infidelity in male faces with accuracy of 4%.

Some other minor points are the following:

What is the scale of the ratings for attractiveness, sexual dimorphism, and trustworthiness?

4% predictability, significant above chance individual rater-level accuracy with only 14.1% to 18.0% of individual raters 14.1% to 18.0% achieved above-chance accuracy in their ratings; are those findings meaningful at all?

Why does the df in table 4 vary? Did you exclude some participants and if yes what the criteria were?

I did not understand the reason behind the analysis reported in table 7, some further clarification is necessary.

Review form: Reviewer 2 (Lynda Boothroyd)

Is the manuscript scientifically sound in its present form?

Yes

Are the interpretations and conclusions justified by the results?

Yes

Is the language acceptable?

Yes

Is it clear how to access all supporting data?

Yes

Do you have any ethical concerns with this paper?

No

Have you any concerns about statistical analyses in this paper?

No

Recommendation?

Accept with minor revision (please list in comments)

Comments to the Author(s)

This is a really nice paper which not only assesses potential accuracy in judgements of sexual behaviour at zero acquaintance by also examines what drives both perceptions and accuracy in perceptions of observers. This is an important step which isn't often taken and really makes this worthwhile.

I have only one slight correction to make re references to our work on perceptions of sociosexuality (Boothroyd et al 2008, 2011) which is cited here as relating to cheating but generally is more about uncommitted sex and number of partners. It's a weakness of the SOI in fact that it doesn't explicitly address cheating.

It's also worth noting that we found more evidence of accuracy in judgements for female faces across our studies than we did for male - the converse of the current results and possibly because of the difference in measure used.

I was really pleased to see nuanced discussion of the effect size and modest individual accuracy rates, as well as discussion of facial cues.

This paper is a lovely addition to the literature.

Decision letter (RSOS-181552.R0)

09-Jan-2019

Dear Dr Foo,

The editors assigned to your paper ("Sexual unfaithfulness can be judged with some accuracy from men's but not women's faces") have now received comments from reviewers. We would like you to revise your paper in accordance with the referee and Associate Editor suggestions which can be found below (not including confidential reports to the Editor). Please note this decision does not guarantee eventual acceptance.

Please submit a copy of your revised paper before 01-Feb-2019. Please note that the revision deadline will expire at 00.00am on this date. If we do not hear from you within this time then it will be assumed that the paper has been withdrawn. In exceptional circumstances, extensions may be possible if agreed with the Editorial Office in advance. We do not allow multiple rounds of revision so we urge you to make every effort to fully address all of the comments at this stage. If deemed necessary by the Editors, your manuscript will be sent back to one or more of the original reviewers for assessment. If the original reviewers are not available, we may invite new reviewers.

When submitting your revised manuscript, you must respond to the comments made by the

referees and upload a file "Response to Referees" in "Section 6 - File Upload". Please use this to document how you have responded to the comments, and the adjustments you have made. In order to expedite the processing of the revised manuscript, please be as specific as possible in your response.

- Data accessibility

If you wish to submit your supporting data or code to Dryad (<http://datadryad.org/>), or modify your current submission to dryad, please use the following link:
<http://datadryad.org/submit?journalID=RSOS&manu=RSOS-181552>

- Competing interests

- Authors' contributions

- Acknowledgements

- Funding statement

on behalf of Dr Bruno Rossion (Associate Editor) and Professor Antonia Hamilton (Subject Editor)
 openscience@royalsociety.org

Associate Editor's comments (Dr Bruno Rossion):

Associate Editor: 1

Comments to the Author:

I received two contrasted reviews. While the second reviews (L. Boothroyd) has only minor points and supports the publications of the manuscript, the first reviewer is more critical, in particular of the null result for female faces and the small effect reported for male faces (accuracy of 4% in judging infidelity from faces). The authors should at least acknowledge and discuss these issues in a revision of their paper, which will be sent to the same reviewer.

Comments to Author:

Reviewers' Comments to Author:

Reviewer: 1

Comments to the Author(s)

The authors examine the accuracy in sexual unfaithfulness judgments of female and male face done by same and opposite sex raters. Both female and male raters were accurate at judging the likelihood of men to cheat and poach but not the likelihood to be unfaithful by females. These judgement seem to be driven by sexual dimorphism, which appears to be a valid cue of infidelity at least for men. The manuscript is well written and overall convincing. Yet, I have two major concerns, which makes me doubt the importance of these findings. The first one is the null result for female faces. Although, it seems plausible that it is more difficult to find cues to infidelity in women's faces, a more reasonable explanation, which the authors also acknowledge, is the limited variability of the self-reported cheating & poaching behavior in women. As the effect sizes are really small of these findings it seems impossible to find any cues to infidelity in female faces. The second major concern is the importance of these findings, it seems to add very little to the already existing literature – males can judge infidelity in male faces with accuracy of 4%. Some other minor points are the following:

What is the scale of the ratings for attractiveness, sexual dimorphism, and trustworthiness?

4% predictability, significant above chance individual rater-level accuracy with only 14.1% to 18.0% of individual raters 14.1% to 18.0% achieved above-chance accuracy in their ratings; are those findings meaningful at all?

Why does the df in table 4 vary? Did you exclude some participants and if yes what the criteria were?

I did not understand the reason behind the analysis reported in table 7, some further clarification is necessary.

Reviewer: 2

Comments to the Author(s)

This is a really nice paper which not only assesses potential accuracy in judgements of sexual behaviour at zero acquaintance by also examines what drives both perceptions and accuracy in perceptions of observers. This is an important step which isn't often taken and really makes this worthwhile.

I have only one slight correction to make re references to our work on perceptions of sociosexuality (Boothroyd et al 2008, 2011) which is cited here as relating to cheating but generally is more about uncommitted sex and number of partners. It's a weakness of the SOI in fact that it doesn't explicitly address cheating.

It's also worth noting that we found more evidence of accuracy in judgements for female faces across our studies than we did for male - the converse of the current results and possibly because of the difference in measure used.

I was really pleased to see nuanced discussion of the effect size and modest individual accuracy rates, as well as discussion of facial cues.

This paper is a lovely addition to the literature.

Author's Response to Decision Letter for (RSOS-181552.R0)

See Appendix A.

RSOS-181552.R1 (Revision)

Review form: Reviewer 1

Is the manuscript scientifically sound in its present form?

Yes

Are the interpretations and conclusions justified by the results?

Yes

Is the language acceptable?

Yes

Is it clear how to access all supporting data?

Not Applicable

Do you have any ethical concerns with this paper?

No

Have you any concerns about statistical analyses in this paper?

No

Recommendation?

Accept as is

Comments to the Author(s)

The authors have revised the manuscript according to my and the other reviewer's comments

Decision letter (RSOS-181552.R1)

21-Mar-2019

Dear Dr Foo,

I am pleased to inform you that your manuscript entitled "Sexual unfaithfulness can be judged with some accuracy from men's but not women's faces" is now accepted for publication in Royal Society Open Science.

on behalf of Dr Bruno Rossion (Associate Editor) and Professor Antonia Hamilton (Subject Editor)
openscience@royalsociety.org

Reviewer comments to Author:

Reviewer: 1

Comments to the Author(s)

The authors have revised the manuscript according to my and the other reviewer's comments

Dr Yong Zhi Foo
School of Psychological Science M304
The University of Western Australia
35 Stirling Highway, Crawley WA 6009
AUSTRALIA

Email yong.foo@uwa.edu.au

Dear Dr Rossion,

Revision of Manuscript (RSOS-181552)

Thank you for the opportunity to revise our manuscript “Sexual unfaithfulness can be judged with some accuracy from men’s but not women’s faces”. We have responded in detail to the reviewers’ comments and made changes to the manuscript.

Please find attached the revised manuscript. Our responses to the reviewers’ comments are detailed below. The reviewers’ comments are shown in italics and our responses are in normal font. Changes to the manuscript are highlighted in yellow.

We are confident that our revised manuscript has addressed your and the reviewers’ points, and is now suitable for publication in *Open Science*.

Yours sincerely,

Dr Yong Zhi Foo (Corresponding Author)
School of Psychology M304,
The University of Western Australia,
35 Stirling Highway, Crawley
WA6009, AUSTRALIA.
Email: yong.foo@uwa.edu.au

Point-by-point reply to comments

Comments from Associate Editor

I received two contrasted reviews. While the second reviews (L. Boothroyd) has only minor points and supports the publications of the manuscript, the first reviewer is more critical, in particular of the null result for female faces and the small effect reported for male faces (accuracy of 4% in judging infidelity from faces). The authors should at least acknowledge and discuss these issues in a revision of their paper, which will be sent to the same reviewer.

We would like to thank both reviewers for their comments. We are very glad that both reviewers have enjoyed reading our MS and found it well-written and convincing. We are also thankful for the opportunity to respond to the comments and further clarify some of the points that we have made. Our detailed comments to the reviewers can be found in the sections below.

Comments from Reviewer 1

The authors examine the accuracy in sexual unfaithfulness judgments of female and male face done by same and opposite sex raters. Both female and male raters were accurate at judging the likelihood of men to cheat and poach but not the likelihood to be unfaithful by females. These judgement seem to be driven by sexual dimorphism, which appears to be a valid cue of infidelity at least for men.

The manuscript is well written and overall convincing.

Thanks!

Yet, I have two major concerns, which makes me doubt the importance of these findings. The first one is the null result for female faces. Although, it seems plausible that it is more difficult to find cues to infidelity in women's faces, a more reasonable explanation, which the authors also acknowledge, is the limited variability of the self-reported cheating & poaching behavior in women. As the effect sizes are really small of these findings it seems impossible to find any cues to infidelity in female faces.

Indeed, the women's self-reported cheating and poaching behaviours in our dataset have limited variance. In particular, the standard deviation of the women's self-reported poaching was at least half of the variance of the men's self-reported cheating and poaching.

Critically, however, the mean and standard deviation of the women's self-reported cheating was comparable to that of men's self-reported poaching (for which we did find some degree of accuracy). Therefore, we think it is unlikely that the lack of accuracy for women's faces is due solely to limited variance in the self-reported behaviours. However, we do agree that it is important to discuss these issues and we have now added a paragraph to our Discussion addressing the issue of limited variability in the self-reported cheating and poaching behaviour of the women's faces (pp. 20-21, line 397-408).

The second major concern is the importance of these findings, it seems to add very little to the already existing literature – males can judge infidelity in male faces with accuracy of 4%.

We believe the finding that men can also rate men's faces has two important implications. The first implication is theoretical. This finding is consistent with the expectation of evolutionary theory that accuracy in our judgments of unfaithfulness from the face might represent an evolved adaptation for detecting not just opposite-sex cheaters, but also same-sex poachers (see pp. 17-18, lines 319-334 in the Discussion).

Second, this result changes our interpretation of previous findings on this topic. As we have mentioned in our Introduction and Discussion, previous papers have only tested accuracy in perception of unfaithfulness from opposite-sex faces. The results from these previous studies appear to show that women could rate men's faces with some accuracy while men could not for women's faces. These findings have led to suggestions that women might be more accurate at perceiving unfaithfulness compared to men. However, our findings here indicate that at least for this face set, the difference is instead due to the sex of the faces rather than the sex of the participants.

We have now re-emphasized these two important points in the concluding paragraph of our Discussion section (pp. 21, line 409-416).

Some other minor points are the following:

What is the scale of the ratings for attractiveness, sexual dimorphism, and trustworthiness?

We have now added the scale of the ratings for attractiveness, sexual dimorphism, and untrustworthiness to our Methods section (pp. 7, line 152-155). We thank the reviewer for noting this omission.

4% predictability, significant above chance individual rater-level accuracy with only 14.1% to 18.0% of individual raters 14.1% to 18.0% achieved above-chance accuracy in their ratings; are those findings meaningful at all?

The relative importance of a 4% effect size depends on the level of explanation sought. A 4% effect size can have substantial evolutionary consequence if that effect is consistently selected for over multiple generations. We have mentioned that our effect size is typical of what is found in evolutionary biology studies. Therefore, the effect sizes in this study can have important evolutionary consequences. We have now added another reference showing that effect sizes that are much smaller than 4% can still have substantial evolutionary impact (pp. 18, line 332).

One point that we did not address in our original manuscript is how our effect sizes compare to other effect sizes in the field of psychology. Importantly, our effect sizes fall within the range of typical effect sizes not just in the field of evolutionary biology, but also in psychology. Several recent studies have shown that effect sizes in psychology averages around $r = 0.2$ (i.e. 4%). Therefore, although the amount of variance shared between rated and actual unfaithfulness is modest, it may still have psychological

relevance. We have now added this information to our Discussion section (pp. 18, line 335-340).

As we have mentioned in our original Discussion, however, despite the potential evolutionary and psychological impact at the group level, we do agree that it is very important to clarify that our results do not mean that individual judgments of unfaithfulness are diagnostic. That is, we are not trying to claim that our results are meaningful in everyday social terms. This point is important to emphasize, especially since several recent papers/commentaries, including our own, have cautioned against interpreting average significant accuracy as evidence of individual diagnosticity (Sutherland et al., 2018; Todorov, Olivola, Dotsch, & Mende-Siedlecki, 2015). We have now expanded on this point to further clarify the difference between average statistical significance and individual diagnosticity in our Discussion (pp. 18, line 341 - 346).

Why does the df in table 4 vary? Did you exclude some participants and if yes what the criteria were?

The individual rater-level negative binomial regression model failed to converge and provide an estimate of the regression slope in 11 instances (two men and three women predicting cheating men's faces, one man and one woman predicting cheating in women's faces, one man and three women predicting women's poaching). These instances were treated as missing data, hence the changes in the dfs. This information was deleted by mistake from an earlier draft of the MS. We have now included it in the Results section (pp. 10, line 214-218) and apologise for this omission.

I did not understand the reason behind the analysis reported in table 7, some further clarification is necessary.

Table 7 was presented to show that the relationship between rated unfaithfulness and actual infidelity in men's faces was accounted for by sexual dimorphism. We have now added the following sentence to clarify the point of the analysis, "If sexual dimorphism was used by raters to make valid judgments of sexual unfaithfulness, then entering sexual dimorphism together with rated unfaithfulness would reduce the relationship between rated unfaithfulness and actual infidelity" (pp. 15 – 16, line 283-286).

We have also changed the description of Table 7 slightly to clarify its purpose. The description now reads. "Results of negative binomial generalized linear models showing that sexual dimorphism accounted for the relationship between ratings of unfaithfulness and actual infidelity, indicating that sexual dimorphism is a driver to accuracy in men's faces" (pp. 16, Table 7).

Comments from Reviewer 2

This is a really nice paper which not only assesses potential accuracy in judgements of sexual behaviour at zero acquaintance by also examines what drives both perceptions and accuracy in perceptions of observers. This is an important step which isn't often taken and really makes this worthwhile.

Thanks!

I have only one slight correction to make re references to our work on perceptions of sociosexuality (Boothroyd et al 2008, 2011) which is cited here as relating to cheating but generally is more about uncommitted sex and number of partners. It's a weakness of the SOI in fact that it doesn't explicitly address cheating.

We have now changed our description of these two references as looking at attitudes towards uncommitted sex and multiple matings (pp. 5, line 97 & pp. 17, line 318).

It's also worth noting that we found more evidence of accuracy in judgements for female faces across our studies than we did for male - the converse of the current results and possibly because of the difference in measure used.

We thank the Reviewer for pointing this difference out. We have now included this point in our Discussion (pp. 19, line 368 - 375).

I was really pleased to see nuanced discussion of the effect size and modest individual accuracy rates, as well as discussion of facial cues.

This paper is a lovely addition to the literature.

We are very pleased with the Reviewer's overall positive review and in particular for noting our careful discussion of the effect sizes.

References

Sutherland, C. A. M., Martin, L. M., Kloth, N., Simmons, L. W., Foo, Y. Z., & Rhodes, G. (2018). Impressions of sexual unfaithfulness and their accuracy show a degree of universality. *PLoS ONE*, *13*, 1–16. doi:10.1371/journal.pone.0205716

Todorov, A., Olivola, C. Y., Dotsch, R., & Mende-Siedlecki, P. (2015). Social Attributions from Faces: Determinants, Consequences, Accuracy, and Functional Significance. *Annual Review of Psychology*, *66*, 519–545. doi:10.1146/annurev-psych-113011-143831